# The Effect of Conjugated Nitrile Structures as Acceptor Moieties on the Photovoltaic Properties of Dye-Sensitized Solar Cells: DFT and TD-DFT Investigation

**DOI:** 10.3390/ijms25137138

**Published:** 2024-06-28

**Authors:** Maha J. Tommalieh, Abdulaziz I. Aljameel, Rageh K. Hussein, Khalled Al-heuseen, Suzan K. Alghamdi, Sharif Abu Alrub

**Affiliations:** 1Physics Department, Faculty of Science, Taibah University, Madinah 44256, Saudi Arabia; mtommalieh@taibahu.edu.sa (M.J.T.); skghamdi@taibahu.edu.sa (S.K.A.); 2Department of Physics, College of Science, Imam Mohammad Ibn Saud Islamic University (IMSIU), Riyadh 11623, Saudi Arabia; aialjameel@imamu.edu.sa (A.I.A.); snabualrub@imamu.edu.sa (S.A.A.); 3Department of Applied Science, Ajloun University College, Al-Balqa Applied University, Ajloun 26873, Jordan; kalhussen@bau.edu.jo; 4Qatar Aeronautical Academy, Doha 4050, Qatar

**Keywords:** nitrile structures, D-A-A organic dye, DFT, TD-DFT, UV–vis absorption spectra, photovoltaic properties

## Abstract

A major challenge in improving the overall efficiency of dye-sensitized solar cells is improving the optoelectronic properties of small molecule acceptors. This work primarily investigated the effects of conjugation in nitriles incorporated as acceptor moieties into a newly designed series of D-A-A dyes. Density functional theory was employed to specifically study how single–double and single–triple conjugation in nitriles alters the optical and electronic properties of these dyes. The Cy-4c dye with a highly conjugated nitrile unit attained the smallest band gap (1.80 eV), even smaller than that of the strong cyanacrylic anchor group (2.07 eV). The dyes lacking conjugation in nitrile groups did not contribute to the LUMO, while LUMOs extended from donors to conjugated nitrile components, facilitating intramolecular charge transfer and causing a strong bind to the film surface. Density of state analysis revealed a considerable impact of conjugated nitrile on the electronic properties of dyes through an effective contribution in the LUMO, exceeding the role of the well-known strong 2,1,3-benzothiadiazole acceptor unit. The excited state properties and the absorption spectra were investigated using time-dependent density functional theory (TD-DFT). Conjugation in the nitrile unit caused the absorption band to broaden, strengthen, and shift toward the near-infrared region. The proposed dyes also showed optimum photovoltaic properties; all dyes possess high light-harvesting efficiency (LHE) values, specifically 96% for the dyes Cy-3b and Cy-4c, which had the most conjugated nitrile moieties. The dyes with higher degrees of conjugation had longer excitation lifetime values, which promote charge transfer by causing steady charge recombination at the interface. These findings may provide new insights into the structure of conjugated nitriles and their function as acceptor moieties in DSSCS, which may lead to the development of extremely effective photosensitizers for solar cells.

## 1. Introduction

Dye-sensitized solar cells (DSSCs) are constantly being investigated in an effort to develop a more economical and environmentally friendly energy source. The main reasons for the extensive exploration of DSSCs are their high performance and reasonable cost as compared to other energy sources [1,2,3]. The advantages of DSSCs based on TiO_2_ films are simplicity of fabrication, low cost, and high efficiency [4,5,6,7]. The most prevalent molecular design for the molecular skeleton of DSSCs is donor–π–acceptor (D–π–A), although there are many different structures as well, including D, A, A, D–A–π–A, D-D–A–π–A, and more. All molecular dye designs have been extensively investigated in order to gain high photovoltaic properties and adsorption stability with the TiO2 surface, thereby attaining a high power conversion efficiency [8,9,10,11]. The high photovoltaic features of these constructed structures require a favorable position of the HOMO and LUMO energy levels to enable dye regeneration and electron injection; for more facilitated intramolecular charge transfer, the dye molecule must be highly conjugated (ICT); and the structure must exhibit a strong absorption band, which corresponds to a high capacity for light harvesting [12,13,14].

The need for one or more chemical substituents, such as distinct acceptor groups, for the dyes in DSSCs to enable their adsorption onto a metal oxide substrate has been confirmed [15,16,17]. The use of such chemical substituents as electron acceptors improves the performance of DSSCs by altering several of their optoelectronic parameters, for example by modifying energy levels and broad-spectrum absorption. An indacenodithiophene-based core was introduced to improve the charge mobility and optoelectronic properties of A-D-A type DSSCs [18]. Studies have been conducted on the effects of the cyano group in the dye’s anchoring section on the stability of its adsorption on the TiO_2_ surface and the other photovoltaic properties, such as open-circuit voltage and the capacity to inject electrons into the surface [19]. Moreover, a number of reviews have addressed the structural and functional aspects of several novel dye anchors for TiO_2_-based DSSCs [20,21,22,23].

Nitriles are a type of organic compound that has a carbon atom (C) bonded to a cyano group (ـــ C ≡ N) in its molecular structure. Nitriles are also referred to as cyanide compounds since their functional groups are cyano. The primary method of producing some nitrile compounds involves heating carboxylic acids and ammonia together with the aid of catalysts. The nitriles that are produced from fats and oils are made using this technique. Heating amides with phosphorous pentoxide is also one of the known methods for producing nitrile compounds [24,25]. The most significant chemical and physical properties of nitriles are their strong dipole–dipole actions, high polarity, and high electronegativity [26]. In addition to the previously mentioned research on the effects of the cyano group as an anchoring group on the performance of DSSCs, other studies have investigated the design of photosensitizers using different cyanide structures in search of effective organic dyes [27,28,29,30].

In this work, D-A-A configurations of DSSC sensitizers were designed with an electron donor bound to an electron acceptor via alkene and nitrile structures in the position of acceptor moieties. The aim was to determine how the photovoltaic performance of the newly designed DSSCs is affected by conjugation in different nitrile components that serve as end acceptors in the D-A-A skeleton. The density functional theory (DFT) and time-dependent DFT (TDDFT) methods were employed as powerful computational tools to gain insight into the key parameters that characterize the overall efficiency of DSSCs, such as electronic structure, molecular energy levels, electron injection, charge regeneration, and efficient light harvesting.

## 2. Results and Discussion

### 2.1. Electronic Structures

The energies of the frontier molecular orbitals (FMOs) are crucial considerations when designing a dye for DSSCs. A well-known technique for studying the FMO energies of molecules to provide an adequate indication of their electronic properties is quantum chemical analysis. The main orbitals addressed with this technique are the highest occupied molecular orbital (HOMO) and the lowest unoccupied molecular orbital (LUMO). The HOMO reveals the electron donor capability, whereas the LUMO describes the capacity to accept an electron. This section covers the main quantum chemical parameters, including the energy gap (Eg), ionization potential (IP = −HOMO), and electronic affinity (EA = −LUMO) [31].

One of the primary parameters used to assess optical properties and provide more accurate details on how sensitizers could affect solar cell performance is the energy gap, or the difference in energy between the HOMO and LUMO. Developing dye molecules with smaller band gaps to facilitate charge transfer was the primary objective of many dye sensitization studies. As seen in Table 1, the calculated values of the energy gaps are found to vary based on the degree of conjugation in the nitrile components. The value of the energy gap is highest in Cy-3a (high level of unconjugated nitrile structure) and smallest in Cy-4c (high level of conjugated nitrile structure). These data indicate that the sequence of conjugation in nitrile components will improve charge mobility in these designed dyes. In fact, the conjugated nitriles were successful in lowering the band gap energy compared with some other well-known anchor groups. The Appendix A gives the chemical parameters that were calculated for the same proposed design but with anchor substituents in the A-site instead of nitrile components. As shown in Appendix A, the strong cyanoacrylic group possessed the smallest band gap (2.065 eV) among the other calculated anchor groups, which was found to be substantially higher than that of Cy-4c (1.80 eV). The ionization potential (IP) is the least amount of energy needed to extract electrons from the HOMO. The minimum amount of energy required to absorb electrons into the LUMO is defined as electronic affinity (EA). Based on the IP and EA values in Table 1, the dye with high regulating conjugation in the nitrile groups has the highest IP and EA values. Therefore, the more consistent conjugation found in the nitrile elements, such as in Cy-2c, Cy-3b, and Cy-4d, made the designed dyes electron-acceptor materials due to their high IP and EA values.

A reliable indicator of intramolecular charge transfer (ICT) is provided by the analysis of the electron density distribution for the frontier molecular orbitals across molecular surfaces. Figure 1 and Figure 2 display the HOMO and LUMO distribution visualizations for the studied dyes. The HOMO has almost the same form for all dyes; its bonding orbitals cover the whole donor TBA, with small lobes on the lower portion of BTD. All dyes exhibit the same distribution of the LUMO on the acceptor BTD, with bonding orbitals located on the benzene ring and antibonding orbitals on the thiadiazol ring. The contribution of nitrile components to the LUMO varies according to their structural compositions. Cy-2a, Cy-2b, Cy-3a, and Cy-4a, which lack single–double and single–triple conjugation, have no contribution to the LUMO. In contrast, the LUMO extended from the BTD to C4, C5, C7, C9, and C10, where the conjugation is observed. The LUMO distribution is observed to be highly uniform in C5, C7, and C10, where conjugation occurs regularly. Notably, conjugation in these dyes enlarged their LUMOs at the expense of the BTD unit; i.e., the LUMO lobes in highly conjugated nitrile are larger than those in BTD. This indicates unequivocally how easily TBA and BTD units can transfer intramolecular charge to the electron-deficient nitrile components in Cy-2d, Cy-3b, and Cy-4c during excitation. Consequently, this results in a strong ability of those dyes to bind to the surface of TiO_2_. The complementarity between the HOMO and LUMO on donor and acceptor units and the ability to separate the HOMO and LUMO are other advantages that are expected to aid in intramolecular charge transfer from donor to acceptor units.

The density of states (DOS) provides the most accurate depiction of the contributions made by the dye’s moieties in the HOMO and LUMO energy levels. Figure 3 and Figure 4 depict the contribution to the total HOMO and LUMO states of each TBA, BTD, and nitrile fragment. For all dyes, the HOMO is more strongly increased by the electron density of the TBA moiety (black color), whereas TBA has the smallest contribution in the LUMO. On the other hand, the BTD made a major contribution (red color) to the formation of the LUMO but had had a small effect on increasing HOMO energy.

Here, the main concern should be how the nitrile structure at the end of each molecule could affect the LUMO energy states. The contribution of the nitrile components to the LUMO is almost negligible in Cy-1, Cy-2a, Cy-2b, Cy-3a, and Cy-4a. A substantial portion of the LUMO is attributed to the nitrile components of Cy-2c, Cy-2d, and Cy-4b. Unexpectedly, C7 and C10 in Cy-3b and Cy-4c were found to have a greater contribution to the LUMO than BTD units (the nitriles C6 and C10 with a green color are taller than BTD with a red color). All of these findings can be attributed to the presence of conjugation in nitrile groups. Due to her severe lack of conjugation, the nitrile structures in Cy-1, Cy-2a, Cy-2b, and Cy-3a had insufficient effects on the LUMO. A partial conjugation in the nitrile structures enhanced their contribution to LUMO energy in Cy-2c and Cy-4b. The full conjugation in the nitrile constituents of Cy-3b and Cy-4c could be the explanation for the cyanides’ dominant contribution to the LUMO relative to the known strong BTD acceptor unit. The conjugated nitrile constituents introduced at the end of the dye contribute most to the formation of the LUMO, which facilitates the transfer of electrons from the HOMO to the LUMO.

HOMO and LUMO energy level order analyses for any organic dye give an adequate indication of the excitation properties and electron transfer capabilities. Figure 5 illustrates the energy level diagram for the designed dyes and references the values of the TiO_2_ conduction band (CB) and the redox level of I^−^/I^3−^. For a dye-sensitized solar cell to be considered efficient, the LUMO levels must remain above the TiO_2_ CB, and the HOMO levels must remain below the redox potential I^−^/I^3−^.

Generally, all dyes display LUMO values higher than the CB edge, indicating a successful electron injection from the excited state (LUMO) to the CB TiO_2_ edge. Also, efficient regeneration is expected for every dye due to the fact that the HOMO energy levels were lower than the redox potential of the electrolyte. The results depicted in Figure 5 indicate that Cy-4c had the lowest energy levels of both HOMO and LUMO, in contrast to Cy-3a, which had the highest levels. This implies that HOMO and LUMO are simultaneously lowered when conjugation is found in nitrile units, while the highest HOMO and LUMO levels result from the absence of conjugation. This will noticeably affect the estimated photovoltaic properties that are calculated later.

### 2.2. UV-vis Absorption Spectra

In order to achieve maximum sunlight harvesting, it is commonly understood that the absorption spectrum of dye sensitizers must include a large portion of the solar spectrum. Therefore, it is desirable to create new sensitizers for organic dyes for DSSCs that have higher absorption coefficients and can absorb in the visible spectrum. Figure 6 displays the simulated absorption spectra of the proposed dyes, and Table 2 provides a summary of maximum wavelength, oscillator strength ƒ (ƒ > 0.4), and electronic transitions. For all dyes, the highest UV-vis absorption maxima are found between 452 and 645 nm. These strong absorption peaks are associated with ICT transfer from HOMO to LUMO electronic transitions, as indicated in Table 2.

The spectra shown in Figure 6 illustrate how the presence of more conjugation in nitrile substituent groups causes the dyes’ absorption bands to be red-shifted. The lowest transition for the first four minimum absorption peaks corresponds to the dyes Cy-3a, Cy-2b, Cy-2a, and Cy-4a, which contain no conjugated nitrile structure. Then, the absorption peaks broaden and become more intense with more redshifts, due to the extended conjugation in nitrile components for the remaining dyes. This is clear in Table 2: the high absorption band peak is associated with a large oscillator strength value. Cy-4c achieved the maximum oscillator strength value, indicating its strong solar radiation harvesting capabilities. Furthermore, the absorption band and oscillator strength of Cy-4c (λ max = 644.94 nm, ƒ = 1.80) are broader and larger than those of Cy-4b (λ max = 550.99 nm, ƒ = 1.43), Cy-2d (λ max = 552.66 nm, ε = 1.53), and Cy-3b (λ max = 583.29 nm, ƒ = 1.45), which can be attributed to Cy-4c’s higher conjugation nitrile segment. This implies that the dye has a greater capacity to absorb solar energy.

### 2.3. Photovoltaic Properties

The performance of solar cells, along with their overall efficiency, is determined by multiple factors. Two primary elements affecting the photocurrent efficiency in DSSCs are the short-circuit photocurrent density (J_SC_) and the open-circuit photovoltage (V_OC_). J_SC_ is affected by both light-harvesting efficiency (LHE) and electronic injection-free energy (ΔG^inject^). LHE is measured by the oscillator strength determined at the maximum absorption wavelength [32].
(1)$$LHE=1-10^{-f}$$

Improved DSSC performance is indicated by higher LHE values. ΔG^inject^ can be used to determine the capacity of electron injection into the conduction band of the semiconductor upon photo-excitation from dyes. The formula for the ΔG^inject^ is as follows [33]:(2)$${\Delta G}^{inject}=E^{dye*}-E_{CB}={(E}^{dye}-E_{00})-E_{CB}$$
where E^dye*^ is the oxidation potential of the excited dye, E_00_ is the lowest excitation energy that corresponds to λ_max_, E^dye^ is the ground-state oxidation potential of dye (E^dye^ = −E_HOMO_), and E_CB_ is the reduction potential of the conduction band edge of TiO_2_, which is commonly expressed as −4.0 eV. Furthermore, the regeneration driving force ΔG^reg^ is used to estimate the dye regeneration efficiency in the excited state, which is an essential consideration for evaluating the performance of DSSCs.

The DSSCs can be made more efficient by using dyes whose E_HOMO_ is closer to the (redox potential I^−^/I^3−^ = −4.8 V). However, the redox agent needs to supply enough driving force for the dye to regenerate effectively. ΔG^reg^ can be found by measuring the difference between E^dye^ and the redox potential [34].
(3)$${\Delta G}^{reg}=E_{redox}-E^{dye}$$

Studies have shown that in order for the electron injection and dye regeneration processes in DSSCs to function effectively, the absolute values of ΔG^inject^ and ΔG^reg^ must be a minimum of 0.2 eV [35]. The open-circuit voltage (V_OC_) serves to evaluate the transfer of electrons from a dye to a semiconductor and can be calculated as follows [36]:(4)$$V_{OC}=E_{LUMO}-E_{CB}$$

The open-circuit voltage value increases with a higher LUMO value (less negative). All of the above-listed parameters that account for the photovoltaic properties of the dyes under investigation are given in Table 3. All dyes exhibit high LHE values within the range of 0.89 to 0.98; Cy-4c was found to have the highest LHE value of 0.98. The optimal photocurrent reactivity could be achieved by all dyes, as indicated by the similar and high LHE values. Since all of the absolute values of ΔG^inject^ were higher than 0.2 eV, all of the dyes revealed adequate driving forces for injecting electrons into TiO_2_. Equation (2) clearly shows that the most negative values of ΔG^inject^ consistently correspond to high HOMO levels. Since the most conjugated nitriles lower the HOMO level (as previously stated), it makes sense that the ΔG^inject^ value for Cy-3a was more negative and the one for Cy-4c was less. Nevertheless, energy redundancy brought on by an overly high value of ΔG^inject^ can result in a smaller V_OC_ [37]. In a similar manner, we will also find that Cy-3a, which has the closest HOMO for redox potential, has the lowest absolute value of regeneration energy ΔG^reg^ (0.56 eV). This can also be explained by the same reason that conjugated nitriles have the lowest HOMO and unconjugated ones have the highest HOMO level, which makes them closer to redox potential. The results of the V_OC_ have a range of 0.26 to 1.01 eV for the proposed molecules, which is adequate for a more effective electron injection from the dye’s E_LUMO_ to the TiO_2_ CB.

The excited-state lifetime (τ) of the material placed between electrodes has considerable effects on the electron transfer in sensitizers. Longer electron lifetimes in an excited state lead to steady charge recombination at the interface, which effectively increases CT within the molecule. The following formula can be used to calculate the excited-state lifetime of organic dyes [38]:(5)$$\tau =\frac{1.499}{f\;E_{ex}^{2}}$$
where f represents the oscillator strength and E_ex_ is the excitation energy. The outcomes of the excited state lifetime are summarized in Table 3. The longer lifetime value of 0.15 ns was recorded for both Cy-2d and Cy-4c, which means the two dyes will have better electron injection into the semiconductor and, consequently, higher overall efficiency.

## 3. Materials and Methods

### 3.1. Chemistry and Molecular Design

Triphenylamine (TBA), a donor unit, is one of the most attractive donor building blocks since it increases the photovoltaic performance of dye molecules when it is included [39]. On the other hand, the low band gap and tunable optical characteristics of 2,1,3-benzothiadiazole ring (BTD)-incorporated D-A dyes have proven to be perfect organic dyes for DSSCs [40]. New organic sensitizers with D-A-A configurations (Figure 7) were designed such that TBA is used as the donor bound to the acceptor unit BTD via alkene, and various nitrile components constituted the second acceptor moiety.

The substituent groups were formed up of nitrile compounds that have one, two, three, or four C≡N functional groups. Figure 8 illustrates the selected nitrile compounds that were collected from a chemical molecule database. Appendix A contains online sources for these compounds.

The designed dyes will be denoted by the symbol Cy, which will be followed by a number indicating how many C≡N units are contained. Figure 9 displays the designed dye configuration, where the base structure (TBA and BTD) is bonded to the different nitrile types. In a prior study, the first dye, Cy-1, was produced experimentally by heating zinc cyanide in NMP at 120 °C along with tetrakis (triphenylphosphine) palladium. The structure of the synthesized molecule was identified during that work using an elemental analysis [41]. The sequence between single–double and single–triple conjugations within the nitrile group is the topic of discussion in the current content. As seen in Figure 9, some nitrile structures lack conjugation, whereas others have semi-conjugated or fully conjugated forms. An extended single–double and single–triple conjugation can be found along the C5, C7, and C10 structural skeletons. Only a partially conjugated configuration exists in C4 and C9. Single bonds (C-C) are dispersed throughout the core of C2, C3, C6, and C8, with no conjugation extending to the terminal C≡N group.

### 3.2. Computational Details

All calculations of both ground and excited states were carried out in the gas phase using the Gaussian 09 quantum chemistry program [42]. The computational model implemented the B3LYP method, which constitutes the combination of the nonlocal Lee–Yang–Parr correlation functional and the nonlocal hybrid exchange functional determined by Becke’s three-parameter method [43,44]. Ground-state geometry optimization was performed using the basis set 6-311 G (d, p) [45]. The Avogadro software 1. 2. 0, an advanced computational chemistry tool, was used to design and visualize the proposed dyes [46]. Density of states (DOS) calculations were performed with the aid of Gauss Sum software [47]. Currently, the TD-DFT approach is widely used as a reliable method to determine the energy of electronic excited states. Furthermore, it has recently been proven that the Coulomb-attenuated hybrid exchange–correlation functional (CAM-B3LYP) is a more accurate model for predicting spectral electronic properties [48]. CAM-B3LYP/6-311G (d, p) was used in the present investigation to calculate the excited state energies and absorption spectra based on the optimized geometry.

## 4. Conclusions

In summary, ten new D-A-A dye-sensitized solar cells that have different nitriles as acceptor moieties were designed. The objective was to explore how the presence of single–double and single–triple conjugation in nitrile structures affected the photovoltaic properties of DSSCs. The optoelectronic and absorption properties of all studied organic dyes were investigated using DFT and TD-DFT. Conjugated nitriles exhibited a smaller energy gap than unconjugated nitriles or other strong anchor groups. Due to their large LUMO contribution, conjugated nitriles are predicted to have high electron mobilities, and no LUMO impact was noted for unconjugated nitriles. The function of conjugated nitriles was most clearly demonstrated by the DOS analysis, which showed that the LUMO was dominated by the conjugated nitrile moieties rather than the popular strong 2,1,3-benzothiadiazole acceptor unit. The energy levels of the proposed sensitizers revealed sufficient thermodynamic stability for electron injection capability (E_LUMO_ > TiO_2_ CB = −4.00 eV) and regeneration through the electrolyte (E_HOMO_ < redox potential = −4.80 eV). The absorption band shifted toward the near-infrared spectrum and became broader and stronger in the presence of conjugation in the nitrile unit. Cy-4c, the most conjugated dye among all the designed structures, has the highest oscillator strength value, the lowest excitation energies, and the highest absorption wavelength, enabling it to operate effectively with the least amount of energy. For all of the studied dyes, better photoelectrical properties are expected due to higher excited-state lifetime and LHE values.

## Figures and Tables

**Figure 1 ijms-25-07138-f001:**
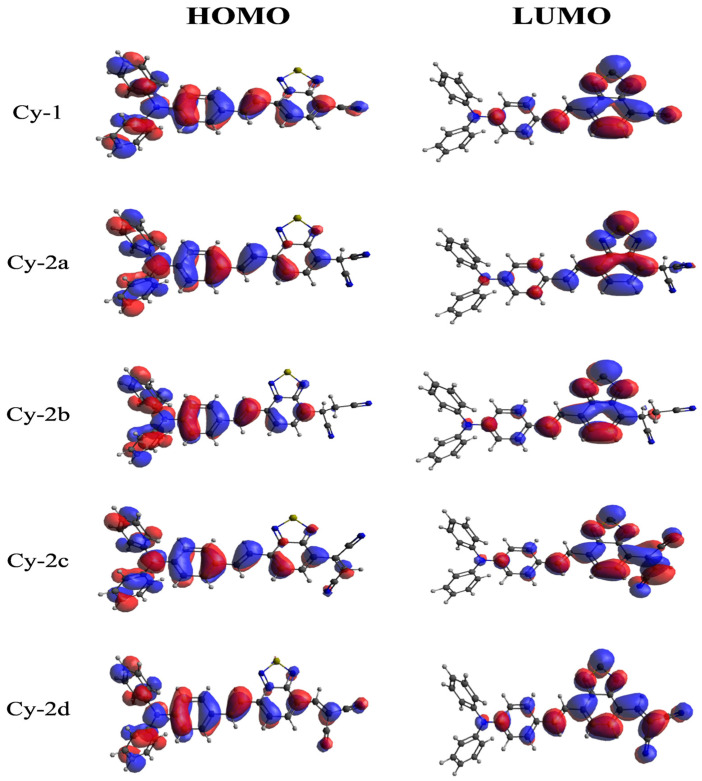
HOMO and LUMO distribution contour map for dyes Cy1→Cy-2d.

**Figure 2 ijms-25-07138-f002:**
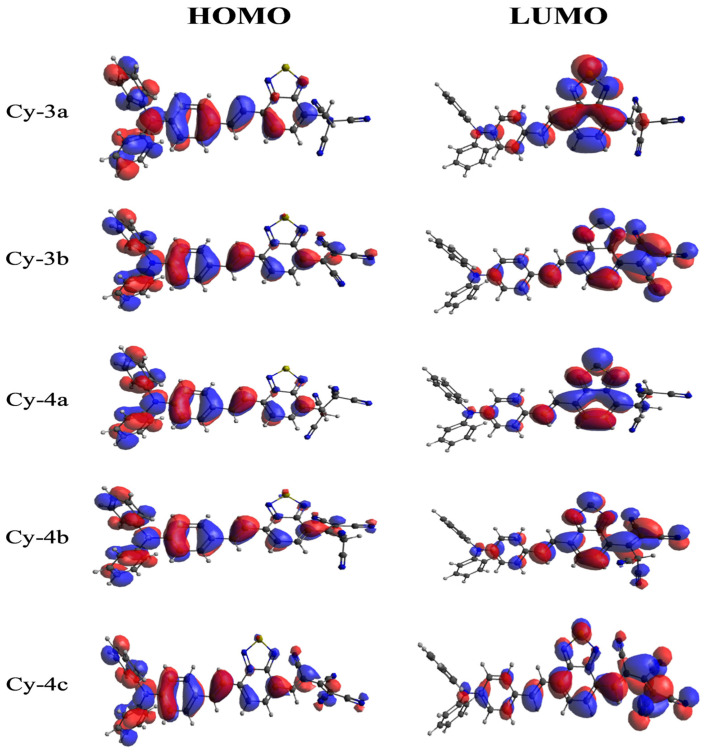
HOMO and LUMO distribution contour map for dyes Cy-3a→Cy-4c.

**Figure 3 ijms-25-07138-f003:**
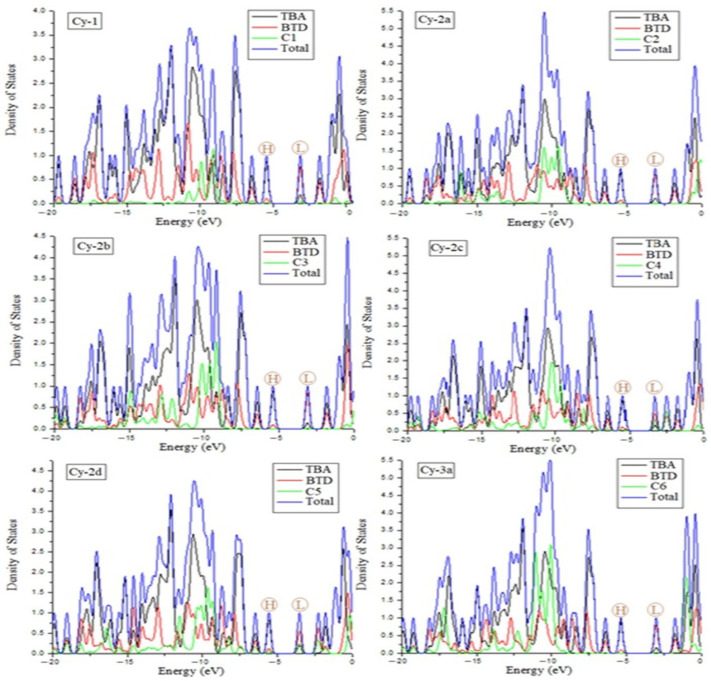
Contributions of donors and acceptors to the density of states (DOS) of the dyes Cy-1→Cy-2d.

**Figure 4 ijms-25-07138-f004:**
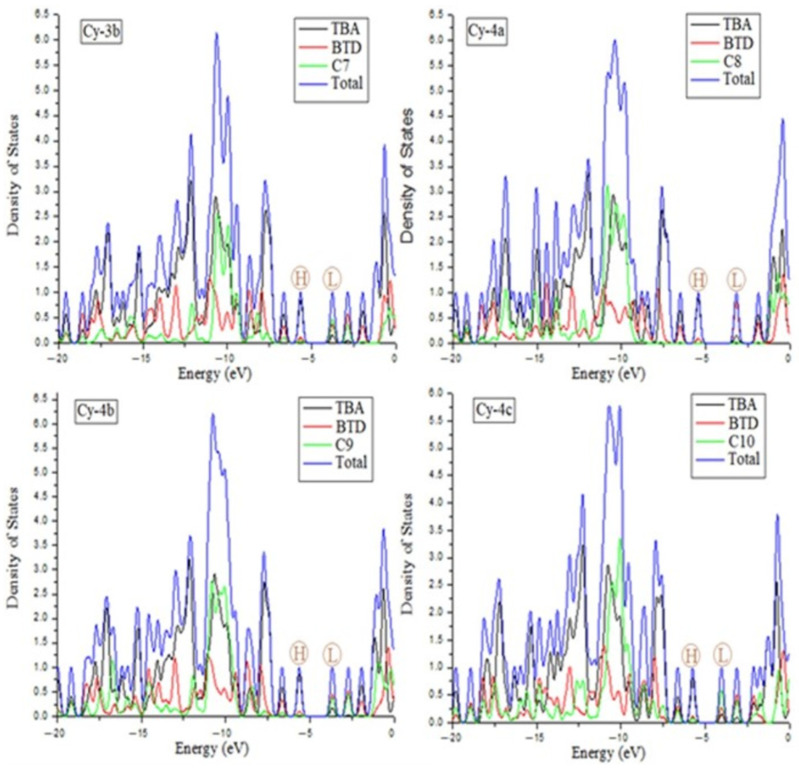
Contributions of donors and acceptors to the density of states (DOS) of the dyes Cy-3b→Cy-4c.

**Figure 5 ijms-25-07138-f005:**
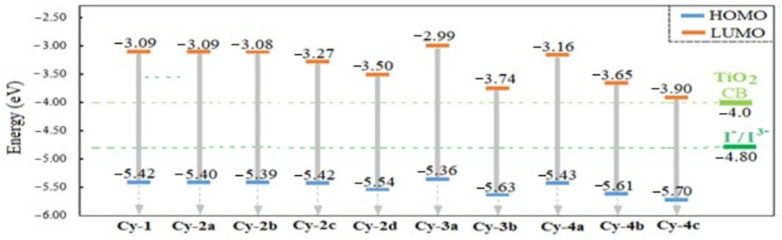
An illustration of the HOMO and LUMO energy levels for each studied dye.

**Figure 6 ijms-25-07138-f006:**
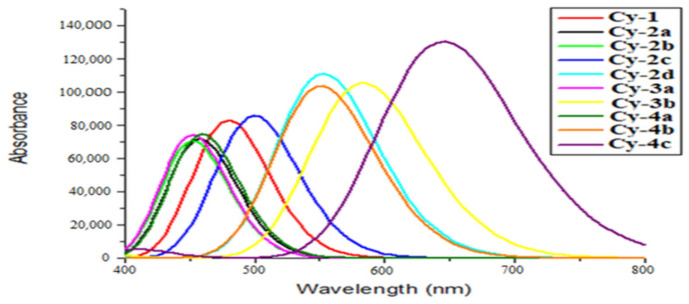
Simulated absorption spectra for all designed dyes.

**Figure 7 ijms-25-07138-f007:**
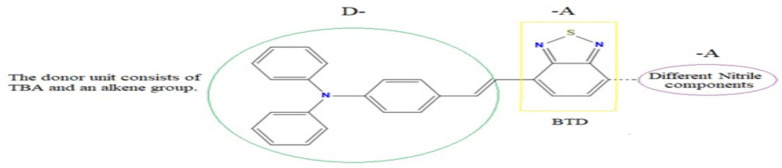
The molecular structure of the designed dye is arranged in a D-A-A configuration.

**Figure 8 ijms-25-07138-f008:**
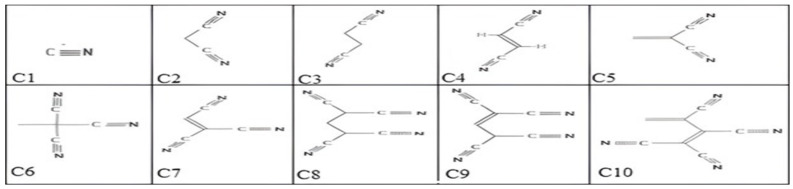
Two-dimensional illustration of the structure of the selected nitrile components.

**Figure 9 ijms-25-07138-f009:**
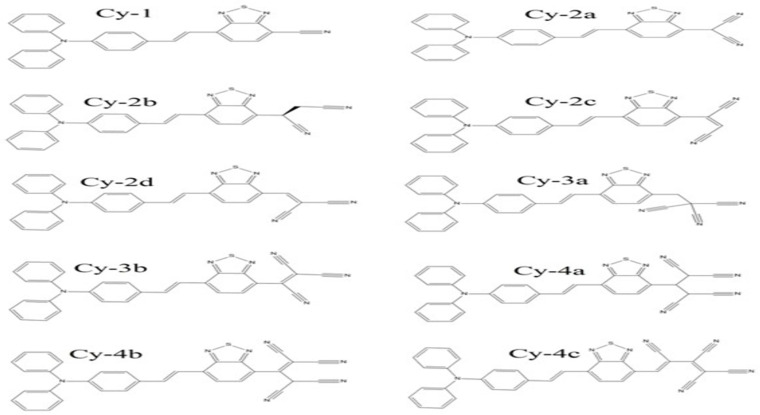
Two-dimensional illustration of the molecular structures of the designed dyes included in the current work.

**Table 1 ijms-25-07138-t001:** The energies of the HOMO and LUMO levels, along with their associated energy gap (Eg), ionization potential (IP), and electronic affinity (EA) values.

Compounds	E_HOMO_ (eV)	E_LUMO_ (eV)	ΔE (eV)	IP (eV)	EA (eV)
Cy-1	−5.42	−3.09	2.33	5.42	3.09
Cy-2a	−5.40	−3.09	2.31	5.40	3.09
Cy-2b	−5.39	−3.08	2.31	5.39	3.08
Cy-2c	−5.42	−3.27	2.15	5.42	3.27
Cy-2d	−5.54	−3.50	2.04	5.54	3.50
Cy-3a	−5.36	−2.99	2.37	5.36	2.99
Cy-3b	−5.63	−3.74	1.89	5.63	3.74
Cy-4a	−5.43	−3.16	2.27	5.43	3.16
Cy-4b	−5.61	−3.65	1.96	5.61	3.65
Cy-4c	−5.70	−3.90	1.80	5.73	3.90

**Table 2 ijms-25-07138-t002:** The electronic absorption properties, including maximum absorption wavelengths, oscillator strength, electronic transitions, and major contributions.

Compounds	E_eX_ (eV)	Wavelength (nm)	Oscillator Strength (ƒ)	Transition	Major Contribution
Cy-1	2.59	479.52	1.15	HOMO→LUMO	77%
Cy-2a	2.71	457.66	0.99	HOMO→LUMO	78%
Cy-2b	2.74	452.19	0.97	HOMO→LUMO	77%
Cy-2c	2.48	499.39	1.18	HOMO→LUMO	77%
Cy-2d	2.24	552.66	1.53	HOMO→LUMO	82%
Cy-3a	2.74	451.84	1.02	HOMO→LUMO	77%
Cy-3b	2.13	583.29	1.45	HOMO→LUMO	79%
Cy-4a	2.70	459.27	1.03	HOMO→LUMO	77%
Cy-4b	2.25	550.99	1.43	HOMO→LUMO	80%
Cy-4c	1.92	644.94	1.80	HOMO→LUMO	75%

**Table 3 ijms-25-07138-t003:** The calculated photovoltaic characteristics of the designed dyes, including injection energy (${\Delta G}^{inject}$), regeneration energy ($\Delta G^{reg}$), light-harvesting efficiency (LHE), open-circuit photovoltage (V_OC_), and excited-state lifetime (τ).

Compounds	E_HOMO_ (eV)	E_LUMO_ (eV)	E_00_ (eV)	E^dyes^ (eV)	E^dyes*^ (eV)	ΔG^inject^ (eV)	ΔG^reg^ (eV)	V_oc_ (V)	LHE	τ (ns)
Cy-1	−5.42	−3.09	2.59	5.42	2.83	−1.17	−0.62	0.91	0.93	0.13
Cy-2a	−5.40	−3.09	2.71	5.4	2.69	−1.31	−0.60	0.91	0.90	0.14
Cy-2b	−5.39	−3.08	2.74	5.39	2.65	−1.35	−0.59	0.92	0.89	0.14
Cy-2c	−5.42	−3.27	2.48	5.42	2.94	−1.06	−0.62	0.73	0.93	0.14
Cy-2d	−5.54	−3.50	2.24	5.54	3.3	−0.70	−0.74	0.50	0.97	0.13
Cy-3a	−5.36	−2.99	2.74	5.36	2.62	−1.38	−0.56	1.01	0.90	0.13
Cy-3b	−5.63	−3.74	2.13	5.63	3.5	−0.50	−0.83	0.26	0.96	0.15
Cy-4a	−5.43	−3.16	2.70	5.43	2.73	−1.27	−0.63	0.84	0.91	0.13
Cy-4b	−5.61	−3.65	2.25	5.61	3.36	−0.64	−0.81	0.35	0.96	0.14
Cy-4c	−5.70	−3.90	1.92	5.73	3.81	−0.22	−0.93	0.10	0.98	0.15

* was to distinguish between the ground-state oxidation potential of dye (symbolized as Edye) and the oxidation potential of the excited dye (symbolized as Edye*).

## Data Availability

The original contributions presented in the study are included in the article, further inquiries can be directed to the corresponding author.

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
