# Peer review of "The Effect of Conjugated Nitrile Structures as Acceptor Moieties on the Photovoltaic Properties of Dye-Sensitized Solar Cells: DFT and TD-DFT Investigation"

_ijms, 2024, doi:10.3390/ijms25137138_

Round 1

Reviewer 1 Report

Comments and Suggestions for Authors

The reviewed manuscript entitled “The Effect of Conjugated Nitrile Structures as Acceptor Moieties on the Photovoltaic Properties of Dye-Sensitized Solar Cells; DFT and TD-DFT Investigation” by Tommalieh et al. represents theoretical study of the group of potential dye-sensitized solar cells based on the D-A-A framework where D= triphenylamine, A = 2,1,3-benzothiadiazole ring (BTD) and second A is the nitrile component serving as second acceptor moiety. The 10 different nitrile compounds were used to create 10 new potential dyes. The DFT and TD-DFT calculations were primarily focused on the electronic structure of studied molecules and their electron transport properties and reactivity (depicted as the HOMO or LUMO-derived properties as ionization potential, to give an example). The density of states (DOS) analysis provided the quantitative description of the contributions made by the investigated terminal nitrile residues in the HOMO and LUMO energy levels. This work is complemented by the analysis of theoretical UV-Vis spectra. The possible electron motion was predicted by the Authors as promising in further utilizing these compounds as solar cells what is additionally supported by the excellent light-harvesting efficiency values which were in range 0.89-0.98.

This paper might be interesting for scientist dealing with the issue of constructing efficient and environmentally friendly energy sources therefore  I recommend to publish this material as it stands.

Author Response

Regards and appreciation to reviewer number one. He has recommended publishing this material as it stands.

Reviewer 2 Report

Comments and Suggestions for Authors

Authors carried out theoretical DFT and TDDFT investigation with Gaussian of a group of photo-sensitive molecules, shown in Figure 3, in total 10 compounds. Using quantum-chemical methods authors calculated ground state properties with DFT (HOMO, LUMO, DOS), as well as excited state properties (excitation energies, oscillator strengths ...) with TDDFT.

Thier results are clearly presented in 3 Tables and 9 Figures.

This is valuable predictive work of new materials for photovolaticity.  The "Cy-4c" as the most conjugated compound is, according to present calculations, the best candidate.  

After careful reading I have only minor issues.

l.45-46 : please provide explanation for the introduced abbreviations D, A ..(donor, acceptor ?)

l.105 : fix "120 o C" to the proper temperature notation of degree of Celsius

Figure 2: make this picture with higher resolution

l.119: please mention in Computational details about applied solvent effects - probably it was vacuum. Question:  the solar cells contain electrolyte. If you would apply solvent effect onto your molecules, how it would affect calculated results ?

l.146 : ad "IP=-HOMO" please provide citation for the statement, Koopmans paper

l. 228 : this is the first mention of the CB abbreviation, please put here the meaning

l.276: for the LHE definition please provide an appropriate reference

l.281: for the deltaG^inject definition please provide an appropriate reference

l.291: for the deltaG^reg definition please provide an appropriate reference

l.296: for the VOC  definition please provide an appropriate reference

l.320: for the excited-state lifetime (tau) definition provide an appropriate reference

l.327: please fix the Table 3 caption ... symbol for injection energy, plus mention that you have also the tau property

l.350: in EHOMO .. use HOMO as small index, like E_{HOMO}.

Author Response

Response to Reviewer #2's comments

(Manuscript: ijms-3065836)

We are very much thankful to the reviewer for his deep and thorough review. We have revised our present research paper in light of his useful suggestions and comments. We hope our revision has improved the paper to a level of satisfaction. Number-wise answers to his specific comments and suggestions are as follows:

Comment 1;

- 45-46: please provide explanation for the introduced abbreviations D, A .(donor, acceptor ?).

Response.

Thank you for the note; Indeed, D and A are abbreviations for donor and acceptor, and an explanation has been provided. 

Comment 2;

- 105: fix "120 o C" to the proper temperature notation of degree of Celsius.

Response. 

As suggested by the reviewer, the proper temperature notation of degree of Celsius has been fixed.

Comment 3;

Figure 2: make this picture with higher resolution.

Response. 

Based on the reviewer's advice, The resolution of Figure 2 has been increased.

Comment 4;

please mention in Computational details about applied solvent effects - probably it was vacuum. Question: the solar cells contain electrolyte. If you would apply solvent effect onto your molecules, how it would affect calculated results?

Response;

The reviewer is right; the calculations have been done in vacuum (gas phase), and this was included in the computational details. This is a valid point; the solvent may have an impact on the optical properties of the studied molecules. The present study was conducted with limited computational capability (a normal computer) on ten molecules. It could take a considerable amount of time to make other calculations. Nevertheless, the point will be carefully considered in future research.

Comment 5;

ad "IP=-HOMO" please provide citation for the statement, Koopmans paper.

Response;

The citation for the statement (IP=-HOMO) has been provided.

Comment 6;

228: this is the first mention of the CB abbreviation, please put here the meaning.

Response;

The meaning for CB abbreviation (conduction band) has been added.

Comment 7;

276: for the LHE definition please provide an appropriate reference.

Response;

The appropriate reference for LHE definition has been provided.

Comment 8;

281: for the ΔG inject definition please provide an appropriate reference.

Response;

The appropriate reference for ΔG inject definition has been provided.

Comment 9;

291: for the ΔG reg definition please provide an appropriate reference.

Response;

The appropriate reference for ΔG reg definition has been provided.

Comment 10;

296: for the VOC definition please provide an appropriate reference.

Response;

The appropriate reference for VOC definition has been provided.

Comment 11;

320: for the excited-state lifetime (tau) definition provide an appropriate reference.

Response;

The appropriate reference for excited-state lifetime (tau) definition has been provided.

Comment 12;

327: please fix the Table 3 caption ... symbol for injection energy, plus mention that you have also the tau property.

Response;

As directed by the reviewer, the Table 3 caption has been fixed, the symbol for injection energy has been corrected, and the tau property has been added.

Comment 13;

350: in EHOMO . use HOMO as small index, like E_ {HOMO}.

Response;

As directed by the reviewer, HOMO in EHOMO has been put as a small index.

Thanks in Advance
